# Shaping of T Cell Functions by Trogocytosis

**DOI:** 10.3390/cells10051155

**Published:** 2021-05-10

**Authors:** Masafumi Nakayama, Arisa Hori, Saori Toyoura, Shin-Ichiro Yamaguchi

**Affiliations:** Laboratory of Immunology and Microbiology, College of Pharmaceutical Sciences, Ritsumeikan University, Shiga 525-8577, Japan; ph0137ie@ed.ritsumei.ac.jp (A.H.); ph0134ek@ed.ritsumei.ac.jp (S.T.); ph0126kk@ed.ritsumei.ac.jp (S.-I.Y.)

**Keywords:** acquisition, nibbling, stripping, cross-dressed, cross-presentation, dendritic cell (DC), TCR, Treg, chimeric antigen receptor (CAR), fratricide, escape variant

## Abstract

Trogocytosis is an active process whereby plasma membrane proteins are transferred from one cell to the other cell in a cell-cell contact-dependent manner. Since the discovery of the intercellular transfer of major histocompatibility complex (MHC) molecules in the 1970s, trogocytosis of MHC molecules between various immune cells has been frequently observed. For instance, antigen-presenting cells (APCs) acquire MHC class I (MHCI) from allografts, tumors, and virally infected cells, and these APCs are subsequently able to prime CD8^+^ T cells without antigen processing via the preformed antigen-MHCI complexes, in a process called cross-dressing. T cells also acquire MHC molecules from APCs or other target cells via the immunological synapse formed at the cell-cell contact area, and this phenomenon impacts T cell activation. Compared with naïve and effector T cells, T regulatory cells have increased trogocytosis activity in order to remove MHC class II and costimulatory molecules from APCs, resulting in the induction of tolerance. Accumulating evidence suggests that trogocytosis shapes T cell functions in cancer, transplantation, and during microbial infections. In this review, we focus on T cell trogocytosis and the related inflammatory diseases.

## 1. Introduction

In order to communicate with each other, immune cells express a wide variety of cell surface molecules such as receptors, ligands, and adhesion molecules. Cell-cell communication is required for the generation of appropriate immune responses to various pathogens. It has been established that during cell-cell interactions, membrane-associated proteins are transferred between immune cells [1,2,3,4]. In the past this biological phenomenon has been called acquisition, nibbling, or stripping, and is currently referred to as trogocytosis, derived from the ancient Greek word *Trogo*, meaning ‘gnaw’ [5]. In contrast to phagocytosis (*Phago* is the Greek meaning ‘to eat’), which is executed by phagocytes such as macrophages and dendritic cells (DCs), trogocytosis is considered to be executed by any type of cells, as described below. By acquiring membrane-associated proteins, so-called *recipient* cells gain alternative cellular functions. In contrast, *donor* cells may lose these proteins and cellular functions (Figure 1). In addition, under certain conditions, bi-directional trogocytosis is observed [6] (see Section 2). In some cases, trogocytosis mediates the intercellular transfer not only of plasma membranes but also of intracellular contents [7,8], which may also alter cellular functions. However, this possibility has not been extensively investigated.

The best characterized trogocytosis involves the transfer of major histocompatibility complex (MHC) molecules from antigen-presenting cells (APCs) to T cells during their interactions [1,3]. Trogocytosis of MHC molecules shapes T cell functions and is involved in various T cell-mediated diseases. Trogocytosis has been observed not only in immune cell interactions, but also during epithelial cell communication [9] and in neuronal synapses [10,11]. Further, trogocytosis is used by amoebae to kill host cells [12,13], indicating that this biological phenomenon is widely conserved throughout eukaryotes. Recent findings of trogocytosis in microbes [14], mammalian neuronal networks [15,16], and non-T cell immune cell interactions [2,17,18] have been well summarized by others previously and also in this issue. Thus, here we mainly focus on T cell trogocytosis and related diseases.

## 2. Possible Mechanisms Underlying Trogocytosis

The molecular mechanisms underlying trogocytosis are not fully understood. However, trogocytosis of the T cell receptor (TCR) and MHC receptor ligand pair has been extensively characterized. Of note, Martinez-Martin et al. have shown that T cells acquire MHC class I (MHCI) from APCs through the action of small GTPases such as RhoG and TC21 [19]. RhoG is known to be involved in phagocytosis [20], and thus trogocytosis is characterized as incomplete phagocytosis. Indeed, PI3K inhibitors effectively inhibit TCR trogocytosis [19,21]. Likewise, natural killer (NK) cells expressing NKG2D, an NK activating receptor, acquire ligands, including RaeI, MICA, and MICB, from tumor cells via a PI3K-dependent pathway [22]. Taken together, trogocytosis appears to be accompanied by recipient cell receptor internalization (Figure 1).

Donor cells may actively release their membrane fragments to recipient cells. These membrane fragments may include extracellular vesicles (EVs) [23]. Choudhuri et al. have shown that TCR signaling leads to secretion of TCR-enriched microvesicles via the central supramolecular activation cluster (c-SMAC) [24]. Sorting of TCR to the c-SMAC and the production of EVs is dependent on tumor susceptibility gene 101 (TSG101), an essential component of the endosomal sorting complex required for transport (ESCRT)-I [25]. These EVs are transferred to and activate neighboring B cells [24]. On the other hand, Kim et al. have reported that TCR signaling causes TCR-enriched microvilli particles, called T cell microvilli particles (TMPs), which are transferred to and activate neighboring DCs [26]. Interestingly, upon TCR stimulation, microvilli provide a structural platform for TCR clustering where TSG101 and the arrestin domain-containing protein 1 (ARRDC1) colocalize, and TCR clusters are released from T cells by the process of trogocytosis [26]. The authors propose that TCR-enriched EVs [24] might also originate from microvilli [27]. Taken together, these studies reveal that, during T-APC interactions, TCR signaling leads not only to MHC trogocytosis but also to secretion of TCR-enriched EVs. By endpoint analysis, this series of events would be viewed as bidirectional trogocytosis. EVs were also recently observed to be released from cytotoxic T lymphocytes (CTLs) to target tumor cells upon TCR activation [28]. Thus, the intercellular transfer of TCRs described below may be mediated by EVs.

In nonimmune cells, TSG101 and ARRDC1 are also involved in formation and release of microvesicles, which are called ARRDC1-mediated microvesicles (ARMMs) [29]. In contrast to exosomes, ARMMs are generated at the plasma membrane [30]. In addition, one cell of two connected cells is reported to engulf the gap junction to take up membrane and cytosol of its neighboring cells, which is called a connexosome [31]. These phenomena may be portions of trogocytosis during nonimmune cell interactions.

It is still unclear how donor-derived proteins exist on recipient cells (Figure 1). The detection of donor proteins on recipient cells by flow cytometry raises the following possibilities. Donor membrane fragments may be merely attached (model A) or fused (model B) to recipient cells. Alternatively, donor-derived proteins may be re-expressed on recipient cells after being internalized and recycled (model C), as proposed in a comment article [32]. Electron microscopy showed that a human NK cell line acquired APC plasma membrane fragments, which were not fused to but rather loosely attached to the plasma membranes of these NK cells [33]. Similar morphology was observed on CD4^+^ T cells that acquired APC plasma membrane fragments (data not shown, but discussed [34]). A study using both a rat T cell subclone synthesizing MHCII and another subclone with acquired MHCII from APCs showed that only the former subclone was sensitive to anti-MHCII antibody-mediated complement lysis. Thus, this suggests the APC-derived membrane fragments are merely attached to T cells [35]. Taken together, these reports support model A (Figure 1) in which donor-derived receptors no longer transmit any signal in recipient cells. In contrast, some studies showed that donor-derived receptors are functional on recipient cells. For instance, in a co-culture of two CD8^+^ T cell clones, one CD8^+^ T cell clone has been observed to acquire TCR from the other T cell clone and subsequently lyse target tumor cells in the acquired TCR-restricted manner, suggesting that the donor TCR is functional on recipient T cells [36]. It was also reported that NK cells acquire CCR7, a chemokine receptor, from donor cells. Further, these NK cells gained migration activity, suggesting that CCR7 is functional on recipient NK cells [37]. These studies support model B or C (Figure 1); however, the intracellular signaling downstream of acquired receptors has not been investigated.

Irrespective of whether model A, B, or C is correct, donor-derived ligands (MHC, costimulatory molecules, etc.) are probably functional on recipient cells. Indeed, numerous studies have shown donor-derived MHC on recipient cells plays important roles in acquired immunity as described in the following sections.

## 3. Tumor

### 3.1. Priming of CD8^+^ T Cells by Cross-Presentation and Cross-Dressing

CTLs play an important role in anti-tumor immunity. To develop anti-tumor CTLs, DCs present tumor antigens with MHCI to naïve CD8^+^ T cells [38,39,40]. DCs are largely divided into conventional DCs (cDC) and plasmacytoid DCs (pDCs) [41]. pDCs are a major producer of type I interferon (IFN-I) in response to viral infection whereas cDCs are the most potent APCs [41,42,43]. The cDCs are further subdivided into DC type 1 cells (DC1s) and DC type 2 cells (DC2s) [41]. DC2s present extracellular antigens on MHCII through the conventional antigen presentation pathway whereas cDC1s are able to present extracellular antigens not only on MHCII, but also on MHCI, called cross-presentation (Figure 2) [39,40,41]. In general, as observed in DC2s, extracellular antigens are processed and loaded on MHCII in phagosomes. In DC1s, however, extracellular antigens are released from phagosomes to the cytosol and then translocated via TAP (transfer associated with antigen processing) molecules to the endoplasmic reticulum (ER) where extracellular antigen peptides as well as intracellular antigen peptides are associated with MHCI [44,45]. Regarding the unusual pathway of extracellular antigens from phagosomes to cytosol, it has been recently reported that DNGR-1 (also known as CLEC9A) senses necrotic cell-derived F-actin [46,47] and its hemITAM-Syk signaling induces phagosomal membrane rupture to allow endocytosed antigens to enter the cytosol in DC1s [48].

In addition to the cross-presentation pathway, several recent studies have reported the cross-dressing pathway, in which DCs acquire MHCI molecules from neighboring DCs or tumor cells (Figure 2). These MHCI-dressed (cross-dressed) DCs activate CD8^+^ T cells via the preformed antigen peptide-MHCI complexes without the above-mentioned antigen processing [3,49,50]. Prior to the first demonstration of the cross-dressing pathway by Dolan et al. [51], the tumor-derived exosomes containing MHCI were previously considered to provoke anti-tumor immunity [52,53]. In this study, when FVB mouse (MHC haplotype: H-2^q^) bone marrow-derived DCs (BMDCs) were co-cultured with dying H-2^b^ tumor cells expressing ovalbumin (OVA), the H-2^q^ BMDCs acquired the OVA peptide-H-2K^b^ complexes from tumor cells and subsequently activated CD8^+^ T cells from OT-I mice specific for OVA residues 257–264 on H-2K^b^. Thus, this indicates that BMDMs do not use self-MHC, but instead use non-self MHC molecules to activate T cells. It should be mentioned here that BMDCs are CD11c^+^ MHCII^+^ cells generated with GM-CSF; however, these *in vitro*-cultured DCs are not equivalent to in vivo DCs and are neither DC1s nor DC2s [54]. To address the role of cross-dressed DCs in vivo, the authors used CD11c-diphtheria toxin receptor (DTR) transgenic BALB/c (H-2^d^) mice in which DCs are removable by diphtheria toxin (DT) treatment [55]. In these mice inoculated subcutaneously with H-2^b^ tumor cells expressing OVA, OT-I CD8^+^ T cells vigorously proliferated, an effect abolished by DT treatment, indicating that DCs are essential for OT-I CD8^+^ T cell proliferation in response to the tumor cell-derived OVA peptide-H-2K^b^ in vivo [51]. Subsequently, cross-dressing has been demonstrated to be involved not only in cancer [56,57,58], but also in transplantation and during microbial infections (see Section 4.1 and Section 5).

Depending on experimental conditions, cross-dressing has been shown to be conducted by both DC1s and DC2s. Further, DC1s are reportedly essential for cross-dressing of DNA vaccine antigens [59,60] whereas DC2s show higher cross-dressing of neighboring DC-derived MHCI [61,62,63]. This apparent discrepancy may be ascribed to the difference in type of donor cells that DCs acquire MHCI from. In addition to cDCs, pDCs also acquire antigen-MHC complexes from tumor cells and stimulate MHC-restricted T cell proliferation [64]. Interestingly, a recent study has shown that pDCs give the antigen-MHCI complexes to DC1s, which contribute to cross-dressing [60].

### 3.2. MHC Trogocytosis in the CTL Effector Phase

Trogocytosis is also frequently observed in the CTL effector phase. When CTLs attack tumor cells, they acquire MHCI from tumor cells (Figure 2) [7,65]. However, it is still under debate whether trogocytosis enhances or suppresses CTL activity. Given the positive correlation between cytotoxic activity and trogocytosis ability [66,67], CTLs with high avidity (high recognition efficiency) may exert both high cytotoxicity and trogocytosis activity. Alternatively, given that the acquired antigen-MHC complexes have been proposed to transmit sustained TCR signals in CD4^+^ T cells [67,68,69,70], trogocytosis may prolong CTL activation. In contrast, a regulatory function of the MHC on CTLs has been also reported. For instance, CTLs that have acquired the tumor antigen-MHC complex are recognized and lysed by tumor-unexperienced CTLs, which is called fratricide cell death (Figure 2) [7,65,71]. Likewise, it was recently reported that trogocytosis-mediated fratricide of chimeric antigen receptor (CAR) T cells causes tumor escape [72] (see Section 3.3).

It is noteworthy that TCR-mediated trogocytosis strips tumor antigens from target tumor cells, causing antigen loss and tumor escape (Figure 2) [2,72,73]. For example, low-avidity CTLs remove tumor antigen-MHCI complexes from target tumor cells without killing, interfering with tumor killing by high-avidity CTLs [73]. Likewise, CAR and monoclonal antibodies (mAbs) also mediate tumor antigen loss via trogocytosis [2,72] (see Section 3.3).

### 3.3. CAR-Mediated Trogocytosis

CARs combine antigen-binding domains, most commonly, a single-chain variable fragment (scFv) derived from the variable domains of antibodies with the signaling domains of the TCRζ chain and additional costimulatory domains from receptors such as CD28, OX40, and 4-1BB [74]. Autologous T cells engineered to express a CAR specific for CD19 (CD19 CAR T cells) are highly effective against several types of B-cell malignancies and have recently received FDA approval for use in children and young adults with relapse of chemotherapy refractory acute lymphoblastic leukemia (ALL) and for adults with chemotherapy-refractory non-Hodgkin lymphoma (NHL) [75]. Despite the high initial response rate with CD19 CAR T cells in ALL, relapse occurs with some tumors being antigen-negative and others antigen-low [75,76,77,78]. A recent study using a mouse model of leukemia demonstrated that CD19 is transferred to CAR T cells via trogocytosis, resulting in removal of the tumor antigen [72]. Such a loss of tumor antigen was also observed during cancer therapies using mAbs such as rituximab and epratuzumab [2,79,80]. This process could cause tumor escape variants. Further, CD19-acquired CAR T cells were shown to be killed by tumor-unexperienced neighbor CAR T cells [72], a process called fratricide (Figure 2) [7]. Therefore, the inhibition of trogocytosis may improve the efficacy of CAR T therapy. Since the specific molecular mechanisms of trogocytosis remain unknown, as an initial strategy, combinatorial targeting could overcome this trogocytosis-based side effect.

## 4. Transplantation

### 4.1. Allospecific T Cell Priming by Cross-Dressing

T cell-mediated recognition of allogeneic transplants has been considered to occur through two main pathways (Figure 3). In the direct pathway recipient T cells recognize intact MHC alloantigens on donor DCs resulting in acute rejection [81]. With the indirect pathway allograft antigens are internalized and processed by recipient DCs and recipient T cells subsequently recognize these antigens, which promotes chronic rejection [82,83]. In addition to these pathways, there is accumulating evidence of a third, semidirect pathway (cross-dressing pathway) where MHC alloantigens are acquired by recipient DCs (Figure 3) [84,85,86] as described below.

In both human and mouse allogeneic DC coculture assays, recipient DCs acquire antigen-MHC complexes from donor DCs, and these donor MHC-dressed recipient DCs prime cognate T cells in a donor MHC-restricted manner, suggesting the role of cross-dressing in T cell alloreactions in vitro [84,87]. In several mouse models of allograft (skin, heart, or kidney) transplantation, recipient DCs infiltrate allografts and acquire donor MHC [88,89]. Finally, these DCs prime alloreactive T cells in a donor MHC-restricted manner, suggesting that cross-dressing indeed occurs in allograft transplantation (Figure 3) [88,89]. However, these studies did not address whether cross-dressed DCs are involved in allograft rejection and which DC subset contributes to the cross-dressing [88].

The relative contribution of cross-presentation and cross-dressing to CD8^+^ T cell activation can be addressed using *TAP*^−/−^ mice. TAP molecules are generally required for cross-presentation, but not for cross-dressing (see Section 3.1). On the other hand, the contribution of DC1s to CD8^+^ T cell activation can be addressed with *Batf3^−/−^* mice, as this transcription factor is required for the development of DC1s, but not of DC2s [90]. Recently, Li et al. used these knockout mice and showed that when H-2K^d^ skin grafts were transplanted into WT or *Batf3*^−/−^ recipient H-2K^b^ mice, *Batf3*^−/−^ recipient mice showed delayed rejection, suggesting that recipient DC1s contribute to allograft rejection [91]. Although DC1s have cross-presenting activity, alloreactive CD8^+^ T cell proliferation was observed in *TAP*^−/−^ mice as well as in WT mice, suggesting that DC1 cross-dressing, rather than cross-presentation, contributes to alloreactive T cell activation [91]. However, it was not directly demonstrated that cross-dressing is involved in allograft rejection. To this end, Hughes et al. used B6 (H-2K^b/b^) WT or *H-2K*^−/−^ recipient mice transplanted with H-2K^b/d^ kidneys expressing the membrane-bound form of OVA; both recipient DCs were found to acquire H-2K^d^ and H-2K^b^-SIINFEKL (OVA-derived peptide) complexes. Two days after transplantation, these mice were adoptively transferred OT-I CD8^+^ T cells. In both recipients, acute rejection was equally observed, indicating that recipient MHCI is not required for rejection. To exclude the possibility of direct pathways (Figure 3), the authors showed that graft survival is prolonged when recipient DCs were depleted using the CD11c-DTR system. Taken together, this study clearly demonstrates that cross-dressed DCs are involved in allograft rejection [92].

In these mouse experiments, recipient DCs acquire allo-MHC from the graft not only via trogocytosis [87,93,94] but also via extracellular vesicles [88,95], although it remains unknown which is the dominant pathway for cross-dressing in transplantation. It also remains unknown whether MHC donor cells in grafts are DCs or parenchymal cells. Furthermore, the most important question concerns whether cross-dressing is essential for allograft rejection because genetically engineered mice in which cross-dressing pathway is specifically impaired have not been developed so far.

### 4.2. Induction of Allospecific T Cell Tolerance by Cross-Dressing

In contrast to skin grafts, allogeneic liver grafts are accepted in mice without any immunosuppressive treatment [96]. In humans, complete immunosuppression withdrawal has proven to be feasible in approximately 20% of liver transplant recipients [97]. These observations led to the hypothesis of spontaneous tolerance in liver transplantation, although the underlying mechanism is not well understood. Ono et al. recently reported that, in a mouse model of allogeneic liver transplantation, recipient DCs infiltrate into liver grafts, and acquire donor MHC. These cross-dressed DCs express high levels of PD-L1, which in vitro did not prime alloreactive CD8^+^ T cells, but rather induced tolerance [98]. Taken together, these results suggest that cross-dressing plays a role in tolerance induction although whether the depletion of PD-L1^high^ cross-dressed DCs causes breakdown of tolerance has not been addressed.

### 4.3. Induction of Allospecific T Cell Tolerance by Double-Negative T (DNT) Cell Trogocytosis

TCRαβ^+^ CD3^+^ CD4^-^ CD8^-^ T cells, so called double-negative T (DNT) cells, comprise a small subset of mature peripheral T cells, and the number of DNT cells are expanded in various inflammatory conditions [99]. Indeed, DNT cells have been reported to be involved in several autoimmune diseases such as systemic lupus erythematosus (SLE), Sjogren’s syndrome, and psoriasis, although the precise origin and function of DNT cells is still under debate [99]. In contrast to such pro-inflammatory activity, a reported regulatory function of DNT cells is the enhancement of allograft survival [100,101,102], in which trogocytosis is involved [100,103]. For instance, in a mouse model of skin allograft transplantation, recipient DNT cells acquire donor MHCI and interact with alloreactive CD8^+^ T cells. During these cell-cell interactions, DNT cells lyse CD8^+^ T cells through the Fas/FasL pathway, which prevents allograft rejection [100,103]. In addition, it has been recently reported that DNT cell trogocytosis suppresses CD4^+^ T cell activation in a mouse model of allergy [104] (see Section 6).

## 5. Infection

Cross-dressing (see Section 3.1) also contributes to antiviral T cell responses, which has been clearly demonstrated by Wakim and Bevan using mouse models of viral infection [62]. In this study, the authors utilized irradiated (H-2K^d^ x H-2K^b^) F1 mice reconstituted with H-2K^d^ CD11c-DTR bone marrow cells, in which DCs have only H-2K^d^ and are removable by DT treatment [62]. Following adoptive transfer of OT-I CD8^+^ T cells and infection with vesicular stomatitis virus expressing OVA, DCs acquired the OVA peptide-H-2K^b^ complex from the virally infected cells. These cross-dressed DCs were essential for memory, but not naïve OT-I CD8^+^ T cell activation, in vivo [62]. Smyth et al. used a mouse model of OVA-expressing adenoviral infection to show that cross-dressing activates not only memory, but also naïve OT-I CD8^+^ T cells [63]. Both studies demonstrated that DC2s have more potent cross-dressing activity than DC1s for antiviral immunity, although they did not use *Batf3*^−/−^ mice [62,63]. The discrepancy regarding cross-dressing of naïve T cells may be ascribed to different amounts of MHCI and costimulatory molecules on cross-dressed DCs. In other words, naive T cells can be primed by DCs with acquired membrane fragments harboring larger amounts of MHCI and costimulatory molecules of virally infected DCs, whereas memory T cells can be activated by DCs dressed with membrane fragments of virally infected parenchymal cells.

In addition to cDCs, pDCs play an important role in immune responses by producing large amount of IFN-I during antiviral immunity (see Section 3.1) [41,43]. Although it is still under debate whether pDCs have antigen processing machinery, pDCs have been reported to have cross-dressing activity [64]. It was also recently reported that pDCs give MHCI to DC1s, which contributes to their cross-dressing [60]. Since these studies measured only CD8^+^ T cell activation, it remains unknown whether direct or indirect cross-dressing by pDCs indeed contributes to antiviral immunity.

## 6. Th2 Diseases

When naive CD4^+^ TCRs recognize antigen-MHCII complexes on APCs, these CD4^+^ T cells expand and differentiate into functionally distinct effector helper T (Th) cell subsets, such as Th1, Th2, and Th17 cells [105]. Among these Th subsets, Th2 cells produce IL-4, IL-5, and IL-13, which play a central role in humoral immunity and host defense against parasite infection, but also have a detrimental role in allergic diseases such as asthma and atopic dermatitis [105]. There are numerous studies showing that naïve CD4^+^ T cells as well as CD8^+^ T cells acquire antigen-MHC complexes from DCs during these cell-cell interactions [68,69,94,106,107,108,109,110]. Upon interaction with DCs, CD4^+^ T cells acquire not only MHCII, but also costimulatory molecules and adhesion molecules that are recruited onto the immunological synapse formed at the cell-cell contact area. Therefore, these MHCII-acquired CD4^+^ T cells are considered to act as APCs [94,106,107,108,109,111,112]. In addition, MHCII acquisition induces prolonged TCR signaling even after dissociation from APCs, which impacts CD4^+^ T cell activation, survival, and cytokine production [70].

In addition to CD4^+^ T cells and DCs, various immune cells acquire MHCII and are involved in Th2 responses. For instance, basophils, the major producer of IL-4 [113], acquire MHCII and act as APCs for Th2 differentiation [114]. Group 2 innate lymphoid cells (ILC2s), which also produce high amounts of Th2 cytokines [115], *per se* synthesize MHCII but also acquire MHCII from DCs and act as APCs in anti-parasitic immunity [116]. 

DNT cells (see Section 4.3) are also involved in allergic asthma. For instance, in a mouse model of OVA-induced allergic asthma, adoptive transfer of DNT cells ameliorates lung inflammation, mucus production, and OVA-specific IgG/IgE production [104]. In this mouse study, DNT cells acquired MHCII molecules from DCs via Lag3/CD223, a CD4 homologue [117] that binds to MHCII. However, it remains unknown how this trogocytosis is involved in suppression of allergic inflammation. Like T regulatory cells (Tregs) [118] (see Section 7), DNT cells may impair the antigen-presenting activity of DCs by stripping off MHCII from their surface. Alternatively, MHCII-acquired DNT cells act as regulatory APCs, such as MHCII-acquired NK cells [119] or lymph node stroma cells [120], which do not express costimulatory molecules and thus induce CD4^+^ T cell tolerance [3]. 

## 7. Treg Trogocytosis

Tregs suppress conventional T cell activation via multiple mechanisms [121,122]. For instance, Tregs absorb IL-2 and produce immunosuppressive cytokines such as IL-10 and TGF-β to inhibit T cell proliferation and function [121,122]. In addition to these direct effects on T cells, Tregs constitutively express CTLA-4 to down-regulate the expression of costimulatory ligands such as CD80 and CD86 on DCs [123]. This extrinsic function of CTLA-4 on Tregs is different from that on effector T cells, in which CTLA-4 transmits the intrinsic inhibitory signal. Treg-specific CTLA-4 deletion indicates that Treg CTLA-4 is crucial for immune suppression [123]. Interestingly, trogocytosis is involved in this process. Specifically, Tregs have been reported to use CTLA-4 to acquire CD80 and CD86 from DCs via trogocytosis (Figure 4) [124,125]. A recent study also reported that induced Tregs (iTregs) have high trogocytosis activity to remove the antigen-MHCII complex from DCs [118]. This activity of iTregs is higher than that of naïve and effector T cells [118], which is probably due to the Tregs form having a more stable immunological synapse (IS) than conventional T cells by excluding protein kinase C-θ (PKC-θ) from the IS [126]. PKC-θ has shown to destabilize the IS [127]. Taken together, trogocytosis may be involved in induction of antigen-specific tolerance by iTregs (Figure 4).

## 8. Application of Trogocytosis

As described above, T cells mediate various diseases such as cancer, autoimmunity, allergy, and infectious diseases. However, in many cases, pathogenic T cells and their TCR antigens remain to be identified, which hampers understanding of pathogenesis and development of therapeutic approaches. To overcome this problem, several approaches for identification of TCR antigens have been developed [128,129,130]. In this context, trogocytosis may also be applied for clinical diagnosis. Specifically, several studies have utilized the ability of CD8^+^ T cells to acquire antigen peptide-MHCI complexes in order to detect antigen-specific T cells in peripheral blood mononuclear cells (PBMCs) from patients infected with human T-cell lymphotropic virus type I (HTLV-1) or lymphocytic choriomeningitis virus (LCMV). Tomaru et al. first successfully identified T cell populations that specifically recognize the HTLV-I Tax (11–19) peptide presented on HLA-A*201 [131]. In this study, the authors established Hmy2.CIR cells, an HLA-A and HLA-B locus-defective immortalized B cell line, transduced with HLA-A*201 fused with GFP. When these cells were cocultured with patient PBMCs, HTLV-I-specific T cells acquired the peptide-HLA-GFP complex and became GFP-positive [131]. This is useful for detection of antigen-specific T cells from bulk PBMCs; however, this approach is limited by cell type, color spectrum of GFP and related proteins, and restriction of each construct to a single MHC. To overcome these limitations, Beadling and Slifka developed a simple and versatile method to detect pathogen-specific T cells called T-cell recognition of APCs by protein transfer (TRAP) assay [132]. Specifically, the authors biotinylated the surface of APCs, followed by labeling with streptavidin-fluorochrome. When cocultured with LCMV-infected APCs labeled with fluorochrome, virus-specific T cells acquired APC membrane fragments and became fluorochrome-positive. Likewise, Daubeuf et al. established a method to detect antigen-specific CD8^+^ T cells by using Dil-labeled APCs [66]. Importantly, this simple method is not limited by type of APC and MHC [132].

In the coculture of T cells and APCs, T cells initially acquire the antigen peptide-MHC complex from APCs and subsequent TCR signaling stimulates the secretion of extracellular vesicles, which are acquired by APCs [26]. As a consequence, the cell-cell contact-dependent intercellular transfer of membrane fragments/vesicles is viewed as bidirectional trogocytosis (see Section 2). By focusing on APCs that acquire T cell membrane fragments containing TCR, Li et al. have recently developed the method to identify TCR ligand [6]. The authors first generated the HLA-A2-restricted single-chain trimer cDNA library containing melanoma neoepitopes and then transduced K562 cells. When cognate TCR recognizes antigen peptides, K562 cells acquire the TCR, which is highly detectable by FACS. After sort-purification, reading of the library-derived antigen sequence enabled identification of neo-tumor antigens [6].

## 9. Conclusions

Trogocytosis has been frequently observed during immune cell interactions and appears to be involved in various diseases. Nevertheless, the molecular mechanisms underlying trogocytosis are still poorly understood. For instance, cross-dressing is involved in CD8^+^ T cell activation in cancer, viral infection, and transplantation; however, it remains unknown how DCs acquire MHCI from donor cells such as other DCs, tumor cells, or virally infected cells. Moreover, how the donor cell-derived MHCI molecules are expressed on the recipient DCs has not been carefully addressed. As a detrimental effect of trogocytosis during the CTL effector phase, the TCR as well as CAR strip off target antigens from tumor cells, resulting in the generation of escape variants. Thus, the inhibition of receptor-mediated trogocytosis may improve the efficacy of some cancer therapies; however, it is currently impossible to inhibit trogocytosis without impairment of receptor functions. It is also unknown how donor cells give their membrane fragments to recipient cells. Understanding of the molecular mechanisms underlying these processes will enable the specific perturbation of trogocytosis pathways, resulting in the development of new therapeutic strategies for treatment of immune diseases.

## Figures and Tables

**Figure 1 cells-10-01155-f001:**
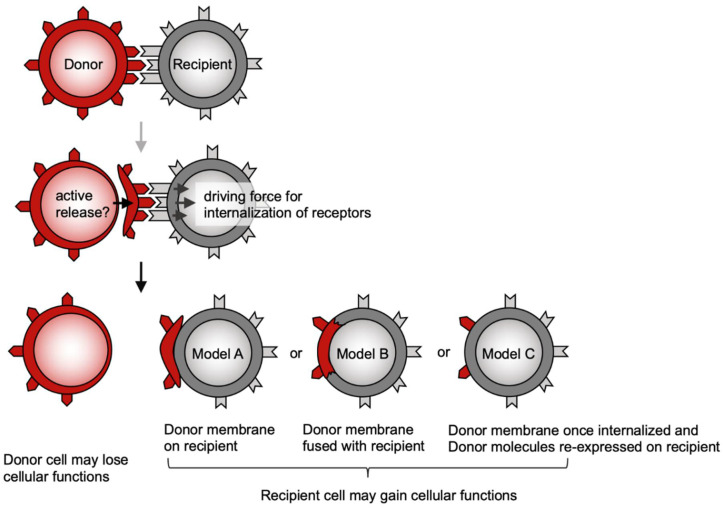
Principal models of trogocytosis. When two types of cells make physical contact via receptor-ligand interactions, the driving force for internalization of receptors expressed on one cell (here called recipient) co-opts ligand-containing plasma membrane fragments from the other cell (here called donor). Not only are receptors and ligands involved in this process, but adhesion molecules also play a role. The donor cells lose membrane molecules and their cellular functions, whereas the recipient cells gain donor-derived membrane molecules and functions. It is unclear how donor-derived molecules exist on recipient cells. The three hypothesized models are shown as A, B, and C.

**Figure 2 cells-10-01155-f002:**
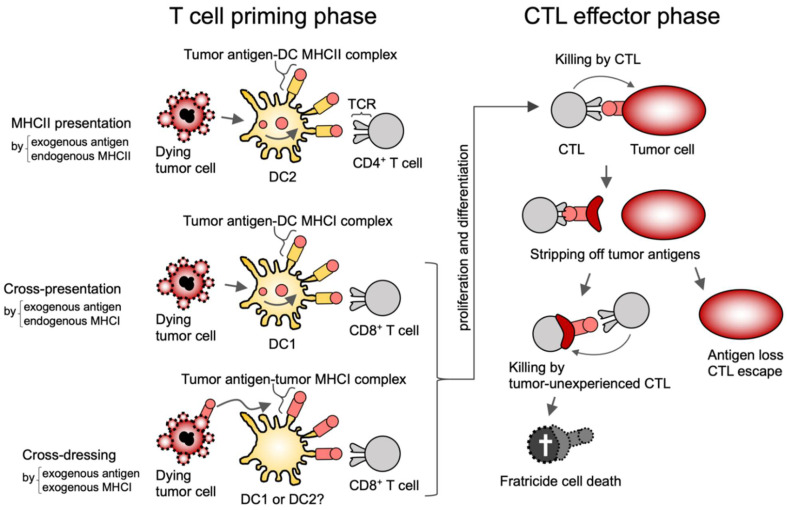
Trogocytosis in T cell priming and effector phases. During the priming phase, dendritic cell (DC) type 2 cells (DC2s) present extracellular tumor antigens on MHCII to activate CD4^+^ T cells whereas DC type 1 cells (DC1s) are able to present them on MHCI, called cross-presentation, to activate CD8^+^ T cells. In addition, DC1s and/or DC2s acquire preformed antigen-MHCI complexes for antigen presentation to CD8^+^ T cells, which is called cross-dressing. In the cytotoxic T lymphocyte (CTL) effector phase, CTLs strip off target antigens from tumor cells. These CTLs with acquired tumor antigen-MHCI are then lysed by tumor-unexperienced CTLs through a process called fratricide cell death. On the other hand, tumor cells lose antigens, resulting in generation of CTL escape variants.

**Figure 3 cells-10-01155-f003:**
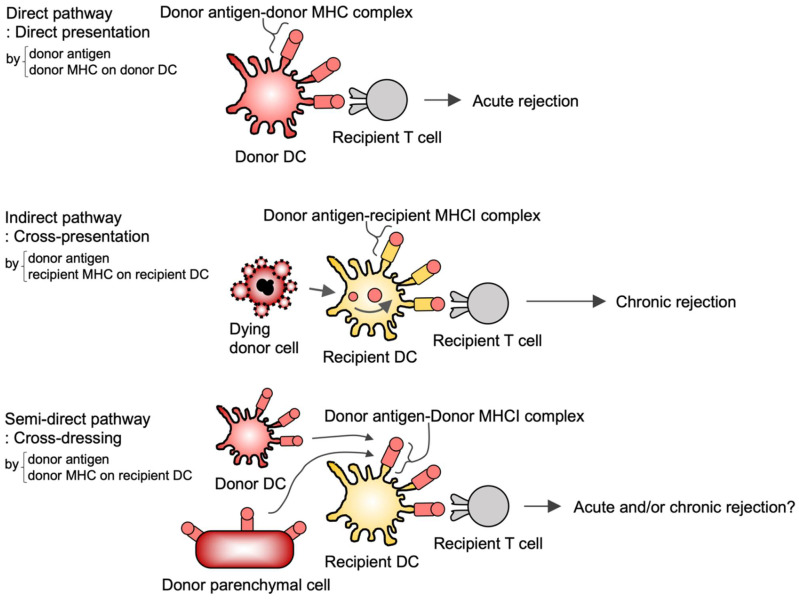
Trogocytosis in allograft transplantation. Alloreactive T cell activation is induced by three pathways. The first is the direct pathway where intact MHC alloantigens on donor DCs are recognized by recipient T cells, promoting acute rejection. The second is the indirect pathway where allograft antigens are internalized and processed by recipient DCs, on which donor antigen-recipient MHC complexes are recognized by recipient T cells, promoting chronic rejection. The third pathway is a semi-direct pathway of so-called *cross-dressing* where recipient DCs acquire preformed donor antigen-MHC complexes and are recognized by recipient T cells.

**Figure 4 cells-10-01155-f004:**
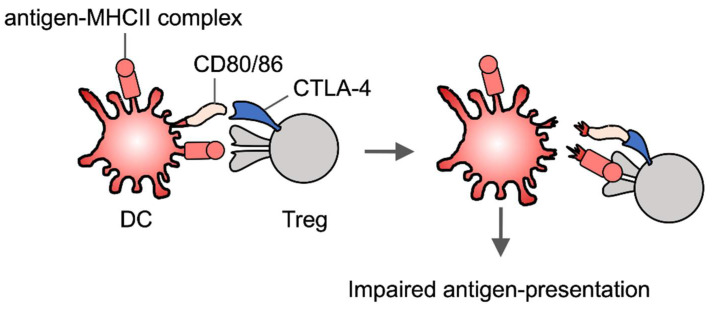
Trogocytosis in Treg-mediated immune suppression. Treg cells strip off MHCII and costimulatory molecules from DCs and as a result these DCs have an impaired antigen-presenting activity.

## Data Availability

Not applicable.

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
