# Peer review of "Shaping of T Cell Functions by Trogocytosis"

_cells, 2021, doi:10.3390/cells10051155_

Round 1

Reviewer 1 Report

The authors nicely summarize recent understandings on trogocytosis, particularly focusing on the modulatory effect of trogocytosis of T cell functions. This reviewer raises only minor comments as described below:

Comment #1.

P2. L6-7 These small GTPases are well-known components of phagocytosis machinery…

                The authors should add appropriate references to this statement. This reviewer agrees that RhoG is reported to be involved in phagocytosis, while involvement of TC21 in phagocytosis seems uncommon.

Comment #2.

P5. L7-8. Alternatively, given that the acquired antigen-MHC complexes have been proposed to transmit sustained TCR signals in T cells [59-62]…

                This reviewer recommends modifying this sentence to “… TCR signals in CD4+ T cells”, since trogocytosis-mediated sustained TCR signaling is mainly observed in CD4 T cells.

Comment #3.

P6. L19-20 DC1 cross-dressing rather than cross-presentation, is essential for alloreactive T cell activation

P6. L29-30 this study clearly demonstrate that cross-dressed DCs are essential for allograft rejection

                These sentences seem to be overstatement that might mislead the readers. Ref #83 only indicates the possible contribution of cross-dressing for alloreactive T cell activation. Ref #84 indicates that cross-dressed DCs are sufficient (and significant) for the induction of allograft rejection, but they cannot demonstrate necessity of cross-dressed DCs for rejection, since engineered mice in which only cross-dressing is impaired have not been developed so far.

Comment #4.

Regarding Figure 4

                It is recommended to add CTLA-4-dependent trogocytosis mechanisms in this Figure, since recently published meeting abstract (Tekguc et al. J Immunol. 2020) also suggest the possible contribution of CTLA-4-mediated trogocytosis in Treg functions. It is also recommended to modify “Loss of antigen-presenting activity”, because antigen presenting activity by DCs does not completely abrogated after the interaction with iTregs.

P6. L24 Typo: SIINEEKL  -> SIINFEKL

Author Response

Thank you very much for reading our manuscript and constructive suggestions. We have now carefully addressed the reviewer’s comments and resubmit the revised manuscript. Point-by-point replies to your comments are as follows:

P2. L6-7 These small GTPases are well-known components of phagocytosis machinery

The authors should add appropriate references to this statement. This reviewer agrees that RhoG is reported to be involved in phagocytosis, while involvement of TC21 in phagocytosis seems uncommon

As per the suggestion of the reviewer, we additionally cited a review paper that summarizes small GTPases in phagocytic machinery (ref #20: Goodridge et al, Traffic, 2012). In addition, as the reviewer pointed out, TC21 has not been demonstrated to be involved in phagocytosis, we have now described “RhoG is known to be involved in phagocytosis” in line #56.

P5. L7-8. Alternatively, given that the acquired antigen-MHC complexes have been proposed to transmit sustained TCR signals in T cells [59-62]…

This reviewer recommends modifying this sentence to “… TCR signals in CD4+ T cells”, since trogocytosis-mediated sustained TCR signaling is mainly observed in CD4 T cells.

As the reviewer suggested, this is mainly in CD4+ T cells. Thus, we have now added “CD4+” in line #163

P6. L19-20 DC1 cross-dressing rather than cross-presentation, is essential for alloreactive T cell activation

P6. L29-30 this study clearly demonstrate that cross-dressed DCs are essential for allograft rejection

These sentences seem to be overstatement that might mislead the readers. Ref #83 only indicates the possible contribution of cross-dressing for alloreactive T cell activation. Ref #84 indicates that cross-dressed DCs are sufficient (and significant) for the induction of allograft rejection, but they cannot demonstrate necessity of cross-dressed DCs for rejection, since engineered mice in which only cross-dressing is impaired have not been developed so far.

We sincerely thank the reviewer for this important comment. We have now understood that these studies do not demonstrate that cross-dressing is “essential” for allograft rejection. Thus, we have now changed “is essential for” to “contributes to” in line #215 or to “involved in” in line #224. Further, we have added the reviewer’s comment on engineered mice in which only cross-dressing is impaired in line #228-230.

Regarding Figure 4

It is recommended to add CTLA-4-dependent trogocytosis mechanisms in this Figure, since recently published meeting abstract (Tekguc et al. J Immunol. 2020) also suggest the possible contribution of CTLA-4-mediated trogocytosis in Treg functions. It is also recommended to modify “Loss of antigen-presenting activity”, because antigen presenting activity by DCs does not completely abrogated after the interaction with iTregs

We also thank the reviewer for this suggestion. We have now added CTLA-4 on Tregs (Figure 4). In addition, as per the suggestion of the reviewer, we changed “Loss of antigen-presenting activity” to “Impaired antigen-presentation” in Figure 4. Although we read the meeting Abstract by Tekguc et al (J Immunol, 204, 1 supplement 2020), we do not cite this as we do not know whether meeting abstract can be cited.

P6. L24 Typo: SIINEEKL  -> SIINFEKL

We thank the reviewer for pointing out the mistype. I have now corrected in line #219.

Reviewer 2 Report

In this review paper, authors focused on T cell trogocytosis and the related inflammatory diseases. They described the potential mechanism of antigen cross-dressing to prime CD8+ T cells. In addition, they described that increased trogocytosis activity is the mechanism of the induction of tolerance in T regulatory cells. Finally, they suggested that trogocytosis shapes T cell functions in cancer, transplantation, and during microbial infections. The review is a good summary of the recent trends in the phenomenon of trogocytosis occurring in cells and its physiological implications.

However, I suggest some revisions to improve the content of this review paper.

1) In this paper, the recent research trends on the effects of trogocytosis on both immune responses and tumors are well summarized, but the mechanism of trogocytosis is not sufficiently addressed.

1-1) Although they described the potential mechanism of trogocytosis in section 2, some parts are misinterpreted. For instance, Dustin group (Ref. 20) reported that T cells release TCR-enriched microvesicles (but not microvilli particles) via the central supramolecular activation cluster (c-SMAC) and suggested that the generation of microvesicles is associated with sorting of TCRs by TSG101. Instead, Jun group (Ref. 21) reported that T-cell microvilli particles (TMPs) are generated in a combined action of trogocytosis and enzymatic cleavage mediated by TCR signaling.

1-2) Although trogocytosis is a well-accepted term in the various fields of cell biology, it is very likely that parts of the membrane and proteins are not purely transferred by trogocytosis. Thus, I suggest to add more information about the potential mechanism of trogocytosis in various biological systems. For example, Kim et al., (Ref 21) demonstrated that TMP formation is not caused by the negative pressure that occurs when two cells fall apart, but the mechanism of microvillar fragmentation and vesiculation.

1-3) I also suggest that the authors refer to the paper of arrestin domain-containing protein 1-mediated microvesicles (ARMMs) (Nature Com 9, 960, 2018) as a potential mechanism for the phenomenon of trogocytosis. The paper shows that ARMMs generate vesicles actively rather than passively on the surface of the cell membrane, as if the viruses are trying to bud on cell membrane.

2) A more detailed explanation is needed as to why T regulatory cells have higher trogocytic activity than other cells. For example, there is a need for arguments about whether T regulatory cells have stronger TCR signals or intracellular signaling system, or whether there is a mechanism by which the level of expression of adhesion molecules such as LFA-1 is increased or activity is increased.

3) Regarding the content of DC1 in Session 4.1, the presence of alloreactive CD8+ T cells in TAP -/- mice is very interesting. However, according to a previously published paper (PMID: 19008445), the alloreactive of CD8+ T cells in TAP-/- mice was significantly reduced. Perhaps, the requirements for a functional TAP may be different depending on the antigen (PMID: 8612129). You can further help readers understand by adding explanations of the above papers to the text.

Author Response

Thank you very much for reading our manuscript and constructive suggestions. We have now carefully addressed the reviewer’s comments and resubmit the revised manuscript. Point-by-point replies to your comments are as follows:

1-1) Although they described the potential mechanism of trogocytosis in section 2, some parts are misinterpreted. For instance, Dustin group (Ref. 20) reported that T cells release TCR-enriched microvesicles (but not microvilli particles) via the central supramolecular activation cluster (c-SMAC) and suggested that the generation of microvesicles is associated with sorting of TCRs by TSG101. Instead, Jun group (Ref. 21) reported that T-cell microvilli particles (TMPs) are generated in a combined action of trogocytosis and enzymatic cleavage mediated by TCR signaling.

We sincerely thank the reviewer for this important comment. We have now understood that Dustin group and Jun group show different vesicles. We have carefully read these 2 papers and discussed in line #62-74.

1-2) Although trogocytosis is a well-accepted term in the various fields of cell biology, it is very likely that parts of the membrane and proteins are not purely transferred by trogocytosis. Thus, I suggest to add more information about the potential mechanism of trogocytosis in various biological systems. For example, Kim et al., (Ref 21) demonstrated that TMP formation is not caused by the negative pressure that occurs when two cells fall apart, but the mechanism of microvillar fragmentation and vesiculation.

As related to above comment, we have now discussed the potential mechanism in line #62-74. As the reviewer suggested, donor cells may provide their membranes in active process, so we added “active process?” in donor cell (Figure 1).

1-3) I also suggest that the authors refer to the paper of arrestin domain-containing protein 1-mediated microvesicles (ARMMs) (Nature Com 9, 960, 2018) as a potential mechanism for the phenomenon of trogocytosis. The paper shows that ARMMs generate vesicles actively rather than passively on the surface of the cell membrane, as if the viruses are trying to bud on cell membrane.

We also thank the reviewer for this suggestion. We have now cited ARMMs paper (ref #29, 30) and discussed in line #78-82. In addition, we have cited a review paper about “connexosome” (ref #31) which may also be a part of trogocytosis.

2) A more detailed explanation is needed as to why T regulatory cells have higher trogocytic activity than other cells. For example, there is a need for arguments about whether T regulatory cells have stronger TCR signals or intracellular signaling system, or whether there is a mechanism by which the level of expression of adhesion molecules such as LFA-1 is increased or activity is increased.

As per this important suggestion by the reviewer, we have now additionally cited 2 papers (ref #126, 127) and discussed the possible mechanism in line #319-321. These papers reported that Tregs form more stable immunological synapse (IS) than conventional T cells by excluding PKC-q from the IS and that PKC-q destabilizes the IS.

3) Regarding the content of DC1 in Session 4.1, the presence of alloreactive CD8+ T cells in TAP -/- mice is very interesting. However, according to a previously published paper (PMID: 19008445), the alloreactive of CD8+ T cells in TAP-/- mice was significantly reduced. Perhaps, the requirements for a functional TAP may be different depending on the antigen (PMID: 8612129). You can further help readers understand by adding explanations of the above papers to the text.

The paper (PMID: 19008445) is Hilder et al. Science 2008. Although we read this paper carefully, we could not find the data using TAP-/- mice. We further sought papers showing the reduced activation of alloreactive CD8 T cells in TAP-/- mice, and we found some (Coelho et al, Transplant Proc, 31, 900-901, 1999; Marrero Suarez et al, Transplant Immunol, 9, 101-110, 2002). However, these studies did not address TAP function in DCs, but focused on the impaired development of CD8+ T cells in TAP-/- mice. Thus, we discuss that TAP-independent antigen-presentation involves cross-dressing in line #208-217.

Reviewer 3 Report

I appreciate authors for needed review about Immune cells trogocytosis. After second spike of Trogocytosis  word in 2000s, understanding of its biology and mechanism of induction, activation and application is biggest challenge. Author's very well written the mechanism of trogocytosis in all possible T cell mediated inflammatory diseases/auto immune diseases. Author's also given potential target of immune disease through trogocytosis.

With interesting reviews, I have some few suggestions,

  1. Author's may be give difference about phagocytosis with Trogocytosis. I know this point is awkward, but I hope it increases readers attention.
  2. Author's only discussed about only membrane protein transport. But recent research showed intra cellular components also transported in trogocytosis. Author's will give short comments in the intra cellular components transportation through trogocytosis and its consequence T cell proliferation and signaling. 

Author Response

Thank you very much for reading our manuscript and constructive suggestions. We have now carefully addressed the reviewer’s comments and resubmit the revised manuscript. Point-by-point replies to your comments are as follows:

Author's may be give difference about phagocytosis with Trogocytosis. I know this point is awkward, but I hope it increases readers attention

As per the suggestion of the reviewer, we have now mentioned the difference between phagocytosis and trogocytosis in line #28-30

Author's only discussed about only membrane protein transport. But recent research showed intra cellular components also transported in trogocytosis. Author's will give short comments in the intra cellular components transportation through trogocytosis and its consequence T cell proliferation and signaling.

We thank the reviewer for this suggestion. We have now cited additional papers showing the transfer of intracellular components via trogocytosis (ref #7, 8), and discussed this issue in line #33-35.